# Children and Adolescents with Early Treated Phenylketonuria: Cognitive Development and Fluctuations of Blood Phenylalanine Levels

**DOI:** 10.3390/ijerph21040431

**Published:** 2024-04-02

**Authors:** Reinhold Feldmann, Ulrike Och, Lisa Sophie Beckmann, Josef Weglage, Frank Rutsch

**Affiliations:** 1Department of General Pediatrics, Münster University Children’s Hospital, 48149 Münster, Germany; ulrike.och@ukmuenster.de (U.O.); frank.rutsch@ukmuenster.de (F.R.); 2Children’s Healthcare Center “Haus Walstedde”, 48317 Drensteinfurt, Germany

**Keywords:** PKU, phenylalanine, blood Phe variations, FSIQ, gain in function

## Abstract

Background: We assessed the relationship between the cognitive development of children and adolescents with phenylketonuria (PKU) and fluctuations in peripheral phenylalanine (Phe) levels. Methods: We examined the neurocognitive performance of 33 children and adolescents with early treated PKU, of whom 18 were treated with sapropterin dihydrochloride, and 15 were on a classic diet. For 26 weeks, patients were assessed weekly for their blood phenylalanine (Phe) levels. Phe levels were analyzed for fluctuations indicated by the individual standard deviation. Fluctuations were compared to the standard deviation of 26 Phe level measurements before the study interval. We also assessed the concurrent IQ of the patients. This was repeated at one-, two-, and seven-year intervals. Results: Full-scale IQ in patients treated with a classic diet did not change within the follow-up. In patients treated with Sapropterin dihydrochloride, however, there was a considerable gain in full-scale IQ. This was particularly true if blood Phe fluctuations increased in patients of this treatment group. Conclusions: Sapropterin dihydrochloride enhances Phe tolerance in patients with PKU. Increasing blood Phe fluctuations following enhanced Phe tolerance may indicate that the treatment not only allows patients to relax their Phe-restricted diet but also may support cognitive development in patients.

## 1. Introduction

Phenylketonuria (PKU), due to phenylalanine hydroxylase (PAH) deficiency, is the most frequent inborn error of amino acid metabolism [1]. The deficient PAH activity results in the inability to convert phenylalanine (Phe) to tyrosine. In patients with classical PKU blood, Phe levels under free nutrition are above 120 μmol/L, leading to concomitantly elevated brain Phe levels. Left untreated, the early rise in blood Phe followed by the accumulation of Phe in the brain is toxic and may lead to severe intellectual and motor disability, eczema, seizures, and developmental and behavioral problems. Thus, treatments aim to decrease blood Phe concentrations.

Patients are identified at birth through newborn screening programs and classified by clinical phenotype. Those with pre-treatment Phe levels > 600 μmol/L require initiation of a Phe-restricted diet. If a diet is initiated shortly after birth that is strictly Phe-reduced and supplemented with a Phe-free amino acid mixture, intellectual and motor disability can be prevented. Subtle intellectual dysfunction, however, is still reported in those with PKU treated early and continuously with the Phe-restricted diet. Patients show a mild reduction in intelligence and neuropsychological deficits. Studies have described reduced attention abilities and slow information processing. Also, severely restricted diets such as the Phe-restricted diet are associated with a risk of nutritional deficiencies, present a substantial psychosocial burden to patients and their families, and pose a challenge with adherence, particularly for adolescents who often fail to comply with the strict recommendations [2,3].

In 1999, a subgroup of PKU patients were identified who responded to exogenous BH4 with reduced blood Phe levels independent of the Phe-restricted diet [4]. Initially thought to be primarily due to higher levels of available BH4, it was later determined to be caused in part by stabilization of PAH protein conformation and by preventing inactivation of the enzyme. In 2009, Sapropterin dihydrochloride (Kuvan™) was approved for the treatment of patients with PKU in combination with a Phe-restricted diet. Sapropterin dihydrochloride (Kuvan™) is increasing PAH activity because of its chaperone effect. This leads to a higher Phe and protein tolerance in responsive patients. Thus, patients who are responsive to Sapropterin dihydrochloride (Kuvan™) are allowed to relax their strict dietary regimen. Prospective and retrospective clinical studies on blood Phe concentrations in sapropterin-responsive patients with PKU showed significant and sustained reductions in blood Phe even if Sapropterin dihydrochloride (Kuvan™) was administered as monotherapy [5,6,7].

European and US protocols exist to determine sapropterin responsiveness in patients with PKU. Protocols first determine the patient’s baseline blood Phe level. In the US, the FDA-approved initial sapropterin dose is 10 mg/kg for one week, and if ≥30% reduction is not achieved, the dose is increased to 20 mg/kg and monitored for a maximum of one month. In the EU, if the Phe level is well controlled, sapropterin treatment is initiated at 20 mg/kg/day. The Phe level is assessed at the 24-h mark. If it has not been reduced by ≥30%, a second dose of sapropterin 20 mg/kg/day is given. The blood Phe level is reassessed at 24 h. In case of a decrease in Phe level < 30% after 48 h of testing, the patient is classified as non-responsive, and sapropterin treatment is discontinued [8].

In individuals with PKU, blood Phe levels are not only elevated but also show greater within-person variability. Most commonly, Phe fluctuations are assessed by the standard deviation of blood Phe (SD) for each subject [9,10,11,12,13,14]. The supplementation with BH4 may decrease variability in blood Phe levels in patients with PKU [15]. The question has arisen whether resulting stability in blood Phe levels is beneficial in children whose blood Phe levels are generally within the recommended range.

The stability of blood Phe levels may be related to cognitive outcomes in early and continuously treated PKU. In a study on children with PKU with at least two blood Phe levels recorded, the correlation between the standard deviations of blood Phe levels and the most recent Full Scale IQ (FSIQ) was −0.36 (*p* = 0.58). This did not reach significance, but it did at least indicate a trend [10]. Indices of metabolic control, such as mean and variation in blood Phe levels, were examined to determine which index best predicted IQ and executive abilities in school-age children with early and continuously treated PKU [13]. Results showed that within the variety of indices of metabolic control, the intraindividual variations in blood Phe levels were a strong predictor of cognitive performance. A study on adults with early treated PKU showed that blood Phe fluctuations, measured by average yearly standard deviations of Phe, correlated negatively with full-scale IQ [16].

Two questions remain substantially unanswered: are fluctuations in metabolic control reducible in young patients with early and continuously treated PKU? If so, is a successful reduction in fluctuations in blood Phe related to a better cognitive outcome in patients with early and continuously treated PKU?

To answer these questions, we performed a study investigating neurocognitive performance in children and adolescents who were treated with a classic diet and sapropterin dihydrochloride. Interestingly, we found an increase in IQ in patients treated with sapropterin dihydrochloride who showed a concomitant increase in blood Phe-level fluctuations.

## 2. Methods

### 2.1. Recruitment

We recruited 54 patients with early and continuously treated PKU. All cases were genetically confirmed by mutation analysis of the PAH gene. Patients were diagnosed with PKU according to their maximal blood Phe level before treatment, which had to be >600 µmol/L to be classified as phenylketonuria. Patients with plasma Phe levels lower than 600 µmol/L were classified as mild hyperphenylalaninemia and did not take part in the current study.

We obtained written consent from these subjects and/or their parents. Two subjects were subsequently excluded from the data analysis due to comorbidities compromising their cognitive functions (birth asphyxia in one case, fetal alcohol syndrome in another); five other formally eligible children’s families did not complete the required blood Phe samplings (see below).

After completing the sampling, every subject was tested with a set of neurocognitive tasks (t1). Tests were repeated after one (t2), two (t3), and seven years (t4). In the follow-up period, there was a considerable drop out of patients. At t3, two more families declined to participate, and four more families had either changed addresses or could not be reached by phone. At t4, six more families declined to participate (all of them referred to their child as a teenager who refused any invitation to participate), and two more families had either changed addresses or could not be reached by phone. Thus, we collected complete data sets of all testing times (t1 to t4) of 33 patients (Table 1).

Over time, the group of patients treated with Sapropterin dihydrochloride (Kuvan™) remained quite stable in size, whereas the group of patients treated with a classic diet decreased in size by some 40%. With regard to age and sex, the patients who dropped out did not differ from those who remained in this study. However, the patients who dropped out had continuously higher blood Phe levels (M 594 μmol/L, SD 336 μmol/L) than patients who remained in this study (M 348 μmol/L, SD 126 mg/dL, *p* = 0.001).

### 2.2. Study Sample

Of the remaining 33 patients, 16 were girls, and 17 were boys. Of the girls, eight were treated with a classic diet, and eight were treated with Sapropterin dihydrochloride (Kuvan™). Of the boys, seven were treated with a classic diet, and ten were treated with Sapropterin dihydrochloride (Kuvan™). At baseline, the patient mean age was 10.5 years (range from 6 to 18 years); at t4, the patient mean age was 17.6 years (range from 12 to 25 years). Treatment groups did not differ significantly regarding age and sex.

### 2.3. Blood Samples

Patients were assessed weekly for their individual blood phenylalanine (Phe) levels. The blood samples needed for individual assessment were always taken at the same time. Parents were asked to report any event that could interfere with Phe levels (such as infection or school excursions). Blood levels of Phe were determined by high-performance liquid chromatography (HPLC) methods following standard procedures. The data obtained in a half-year span (26 assessments) were analyzed for fluctuations indicated by the standard deviation of the individual blood Phe levels. Fluctuations were compared to the standard deviation of 26 individual blood Phe level measurements collected by monthly clinical routine before baseline. This was performed for all patients. At baseline, responsive patients started the sapropterin treatment, and non-responsive patients continued their classic diet.

### 2.4. Psychometric Tests

The test battery included the Wechsler Intelligence Scales for children (WISC-IV) and for adults (WAIS-IV). The WISC-IV can be administered to children aged 6 to 15 years, whereas the WAIS-IV is standardized for individuals aged 16 and older [17,18]. Both tests are considered classic tools to assess general intelligence as well as four subcategories (“index scores”), including verbal comprehension, fluid reasoning, working memory, and processing speed.

Informed written consent was obtained in all cases after the nature and possible consequences of this study were fully explained. Consent to the study protocol was given by the local ethics committee. Bivariate Pearson correlations and Repeated Measures ANOVA were carried out using IBM SPSS Statistics version 25.

## 3. Results

### 3.1. Blood Phe Levels and Blood Phe Fluctuations

Blood Phe levels in patients (total sample) increased moderately from pre-study measurement to study measurement. Blood Phe levels did not differ between treatment groups. Fluctuations, however, were lower in the patients treated with Sapropterin dihydrochloride (Kuvan™). In both treatment groups, fluctuations decreased from pre-study measurement to study measurement (Table 2).

Blood Phe level fluctuations in patients treated with a classic diet were closely related to the Blood Phe levels themselves (Table 3).

However, there was no close correlation between blood Phe levels and their fluctuations in patients treated with Sapropterin dihydrochloride (Kuvan™, Table 4).

Mean blood Phe tolerance in patients treated with a classic diet was 431.4 μmol/L (SD 112.1 μmol/L). In patients treated with Sapropterin dihydrochloride (Kuvan™), mean blood Phe tolerance before treatment was 688.7 μmol/L (SD 419.6 μmol/L, *p* = 0.037). In patients treated with Sapropterin dihydrochloride (Kuvan™), mean blood Phe tolerance increased during therapy to 1470.0 μmol/L (SD 790.8 μmol/L).

### 3.2. Cognitive Development

Full-scale IQ in patients treated with a classic diet did not change within the seven-year follow-up. In patients treated with Sapropterin dihydrochloride (Kuvan™), however, there was a considerable gain in full-scale IQ (Table 5).

Regarding the index scores of WISC-IV and WAIS-IV, both treatment groups showed minor improvement in verbal comprehension, fluid reasoning, and working memory. In patients treated with Sapropterin dihydrochloride (Kuvan™), however, there was a considerable gain in processing speed (Table 6).

Patients treated with a classic diet had a minor FSIQ gain in the seven-year follow-up if blood Phe fluctuations decreased. This was true for six of 15 patients. If blood Phe fluctuations increased, patients treated with a classic diet had a minor loss of FSIQ. This was true for nine of 15 patients.

Patients treated with Sapropterin dihydrochloride (Kuvan™) had a minor FSIQ gain in the seven-year follow-up if blood Phe fluctuations decreased. This was true for eight of 18 patients. If blood Phe fluctuations increased, patients treated with Sapropterin dihydrochloride (Kuvan™) had a considerable gain in mean FSIQ. This was true for ten of 18 patients (Table 7).

In patients treated with Sapropterin dihydrochloride (Kuvan™), an increase in blood Phe fluctuations was true for patients with comparatively high Blood Phe levels (M 7.3 mg/dL, SD 2.0 mg/dL); a decrease in blood Phe fluctuations was true for patients with comparatively low Blood Phe levels (M 5.9 mg/dL, SD 1.3 mg/dL, *p* = 0.129). Patients with an increase in blood Phe fluctuations had a lower mean blood Phe tolerance before treatment (631.3 μmol/L, SD 450.4 μmol/L) than patients with a decrease in blood Phe fluctuations (722.5 μmol/L, SD 386.4 μmol/L, *p* = 0.67). Patients with an increase in blood Phe fluctuations showed a lower gain in mean blood Phe tolerance during treatment (743.8 μmol/L, SD 650.0 μmol/L) than patients with a decrease in blood Phe fluctuations (846.3 μmol/L, SD 443.8 μmol/L, *p* = 0.72).

## 4. Discussion

### 4.1. Lowering Phe Fluctuations

In both treatment groups, the mean fluctuations in blood Phe levels decreased slightly. The decrease, thus, cannot be attributed to only one therapy (classic dietary treatment vs. treatment with Sapropterin dihydrochloride). An observation bias, however, could account for this finding. PKU patients, who knew that their blood Phe concentrations were closely monitored under study conditions, may have shown a more continuous adherence to dietary recommendations over the study phase. Some authors suggest that testing frequency can be a useful predictor of adequate metabolic control and that more regular Phe samples are associated with overall lower median Phe concentrations [11]. Weglage et al. (1992) reported that about four out of five patients indicated greater dietary adherence directly preceding Phe level assessments [19].

### 4.2. Impact of Blood Phe Fluctuations

Few studies have assessed the impact of blood Phe variability on neurocognitive function. In 1998, Arnold et al. described associations of blood Phe variability with executive functions but not with cognition or motor skills [9]. Hood et al. (2014) reported that blood Phe variability predicted cognitive outcomes more strongly than the mean Phe level or the index of dietary control [13]. They also cited a study by Vilaseca et al. (2010), a paper that focused on the impact of dietary control on general intelligence without discussion of particular neurocognitive tests and domains [20]. Similar to Hood et al., Romani and colleagues (2017) listed a number of correlations between Phe fluctuations and various cognitive measures [16]. In contrast, Viau et al. (2011) found no relationship between Phe fluctuations and cognitive performance [11]. An earlier study by Anastasoaie et al. (2008) suggested a potential relationship between variability in Phe and IQ, reporting a trend without statistical significance; neurocognitive function was not investigated in that study [10].

In our study, blood Phe fluctuations had no impact on FSIQ development in patients treated with a classic diet. However, in patients treated with Sapropterin dihydrochloride (Kuvan™), mean FSIQ improved with increasing blood Phe level fluctuations. This observation seems to be in contrast to previous studies, which claimed that high blood Phe level fluctuations are detrimental to cognitive performance in patients with PKU. However, in our study, even increased Phe fluctuations were still within or close to the target range.

According to our results, blood Phe level fluctuations are not threatening for some patients with PKU. These patients had moderate PKU, were treated with Sapropterin dihydrochloride (Kuvan™), and showed a comparatively lower blood Phe tolerance before treatment. Their increasing blood Phe level fluctuations are correlated to a gain in FSIQ that can be traced back primarily to a speed gain in information processing. Interestingly, this was seen in patients whose processing speed before treatment was poor.

The considerable increase in mean FSIQ in patients treated with Sapropterin dihydrochloride (Kuvan™) following increasing blood Phe level fluctuations is difficult to discuss. Sapropterin dihydrochloride (Kuvan™) enhances Phe tolerance in patients with PKU. Patients, thus, may relax their Phe-restricted diet and are ‘allowed’ to exist on a normal or close to normal diet. A Phe-restricted diet with amino acid supplementation is a true blessing for patients with PKU. However, a normal diet may still be in favor if normal cognitive development is considered. Thus, Sapropterin dihydrochloride (Kuvan™) may not have a direct impact on cognitive development in treated patients. It seems, however, to positively influence cognitive development in patients by allowing them to enjoy a normal or close-to-normal dietary life.

In patients treated with Sapropterin dihydrochloride (Kuvan™), considerable gain in IQ was particularly found in those patients who had a lower blood Phe tolerance (before and during therapy) and comparatively higher blood Phe levels (during therapy). These patients may have relaxed their Phe-restricted diet more than those patients with comparatively lower blood Phe levels. They, thus, may have an even higher benefit from diet change.

If so, what is the role blood Phe fluctuations play in FSIQ alterations? First of all, blood Phe fluctuations may not be as detrimental to cognitive development as has been stated before. Then again, changes in blood Phe fluctuations seem to be a result of blood Phe level changes in patients treated with Sapropterin dihydrochloride (Kuvan™). Increasing blood Phe fluctuations, thus, may indicate a successful treatment that not only allows for the diet to be relaxed but also leads to remediation of FSIQ in patients.

### 4.3. Limitations

The limitations of this study should be acknowledged. Follow-up compliance was poor in patients treated with a classic diet. With regard to age and sex, the patients who dropped out did not differ from those who remained in this study. The single-sided dropout may be due to a different compliance of patients who presumably feel successfully treated (with Sapropterin dihydrochloride) compared to patients who may just feel controlled (metabolic control via classic diet) and missing a continuous personal benefit from study adherence.

On the other hand, the patients who dropped out had higher blood Phe levels than patients who remained in this study. Presumably, patients who dropped out had lower overall compliance, which affected not only their adherence to diet but also their adherence to this study.

At the time of recruitment, patients’ age varied from five to 18 years. Similar to what Trefz et al. (2011) discuss regarding the assessment of treatment success, our sample heterogeneity may limit conclusions about the impact of Phe fluctuations on clinical outcomes in various age groups [15]. Each age group may require a different set of neurocognitive tests, but the definition of age groups often depends on practical issues of sample selection and less on neuropsychologically justified age discrimination [21,22]. As van Spronsen et al. (2011) point out, monitoring neurocognitive functioning in PKU patients requires the same conceptually informed instruments for different age groups [23].

The intergroup comparisons of patients treated with Sapropterin dihydrochloride (Kuvan™) and patients treated with classic diet might have suffered from a selection bias as patients treated with Sapropterin dihydrochloride (Kuvan™) tend to have milder phenotypes given their residual enzyme function. However, this bias does not affect the conclusions regarding the relationship between overall cognitive performance and blood Phe level fluctuations.

## 5. Conclusions

Sapropterin dihydrochloride enhances Phe tolerance in patients with PKU. Patients are allowed to relax their Phe-restricted diet and to exist on a normal or close-to-normal diet. Thus, Sapropterin dihydrochloride may positively influence cognitive development in patients by facilitating a normal dietary life. Blood Phe fluctuations may not be as detrimental to cognitive development as has been stated before. If anything, increasing blood Phe fluctuations may indicate a successful treatment that not only allows for the diet to be relaxed but also leads to remediation of FSIQ in patients.

## Figures and Tables

**Table 1 ijerph-21-00431-t001:** Number of patients tested.

Testing Time	Number of Patients Tested (Classic Diet, Sapropterin)
t1 (baseline, 2013)	47 (26, 21)
t2 (one-year follow-up, 2014)	47 (26, 21)
t3 (two-year follow-up, 2015)	41 (23, 18)
t4 (seven-year follow-up, 2020)	33 (15, 18)

**Table 2 ijerph-21-00431-t002:** Blood Phe levels and blood Phe fluctuations (μmol/L).

Phe (mg/dL)	Total Sample	Classic Diet	Sapropterin
Mean Blood Phe level pre	342 (SD 120)	348 (SD 162)	336 (SD 90)
Mean Blood Phe level post	396 (SD 186)	402 (SD 264)	396 (SD 96)
Fluctuations in Phe levels pre	156 (SD 66)	186 (SD 78)	138 (SD 42)
Fluctuations in Phe levels post	120 (SD 48)	144 (SD 48)	102 (SD 42)

Pre: pre-study measurement; post: study measurement.

**Table 3 ijerph-21-00431-t003:** Blood Phe levels and blood Phe fluctuations in patients treated with classic diet.

Blood Phe Level	Fluctuations in Phe Levels Pre	Fluctuations in Phe Levels Post
Mean Blood Phe level pre	0.783 **	0.644 **
Mean Blood Phe level post	0.741 **	0.431

Pre: pre-study measurement; post: study measurement, ** *p* < 0.01.

**Table 4 ijerph-21-00431-t004:** Blood Phe levels and blood Phe fluctuations in patients treated with Sapropterin dihydrochloride (Kuvan™).

Blood Phe Level	Fluctuations in Phe Levels Pre	Fluctuations in Phe Levels Post
Mean Blood Phe level pre	0.330	0.433
Mean Blood Phe level post	−0.176.	0.269

Pre: pre-study measurement; post: study measurement.

**Table 5 ijerph-21-00431-t005:** FSIQ development in patients treated with classic diet or with Sapropterin dihydrochloride (Kuvan™).

FSIQ	Classic Diet	Sapropterin
t1	103.0 (SD 6.9)	95.3 (SD 13.6) *
T2	105.0 (SD 9.4)	98.9 (SD 12.8) *
T3	104.0 (SD 9.5)	100.8 (SD 13.6) *
T4	103.5 (SD 6.9)	101.6 (16.3) *

* *p* < 0.05 (repeated measures).

**Table 6 ijerph-21-00431-t006:** Processing speed development in patients treated with classic diet or with Sapropterin dihydrochloride (Kuvan™).

Speed	Classic Diet	Sapropterin
t1	100.3 (SD 12.1)	89.9 (SD 10.7) *
T2	104.7 (SD 13.9)	96.1 (SD 11.8) *
T3	103.6 (SD 11.0)	95.6 (SD 10.5) *
T4	101.3 (SD 14.2)	98.0 (16.5) *

* *p* < 0.05; (repeated measures).

**Table 7 ijerph-21-00431-t007:** Development of blood Phe level fluctuations and FSIQ during seven-year follow-up in patients treated with classic diet or with Sapropterin dihydrochloride (Kuvan™).

FSIQ Development	Classic Diet	Sapropterin
Blood Phe fluctuations decreased	2.4 (SD 11.0)	1.7 (SD 6.9)
Blood Phe fluctuations increased	−1.2 (SD 5.1)	9.25 (SD 5.9) *

* *p* < 0.05 (repeated measures).

## Data Availability

The data presented in this study are available on request from the corresponding author.

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
