# Peer review of "Children and Adolescents with Early Treated Phenylketonuria: Cognitive Development and Fluctuations of Blood Phenylalanine Levels"

_ijerph, 2024, doi:10.3390/ijerph21040431_

Round 1

Reviewer 1 Report

Comments and Suggestions for Authors

With great interest I have read the manuscript of Feldmann et al., entitled “Children and adolescents with early treated phenylketonuria: cognitive development and fluctuations of blood phenylalanine levels”.

I believe that the manuscript can strengthen it by providing additional information on the following issues:

Introduction

The study starts with a general review of the literature reporting many studies on metabolic control in PKU but not mentioning recent papers on PKU and cognitive functions in children and adolescent (see Leuzzi et al. (2014, Age-related psychophysiological vulnerability to phenylalanine in phenylketonuria; Hawks et al., 2018, Developmental Trajectories of Executive and Verbal Processes in Children with Phenylketonuria; Manti et al., 2017, Predictability and inconsistencies in the cognitive outcome of early treated PKU patients).

Page 1

Line 33 and line 34

“…mental retardation.”

Rephrase with “..intellectual disability..”

Page 2

Line 56

p= .058

Page 3

Line 96 and line 97

It would be better to convert phe level in µmol/L

Page 3

2.2. Study sample

…”At baseline, patient mean age was 10.5 years 102 (range 5 to 18 years)….

What kind of instrument has been used for assessing the IQ in the patients 5 years old?

In the method section this information is lacked.

Page 3

2.3. Blood samples

The paragraph on blood samples is difficult to understand. I would suggest reworking this section.

My main concern is that data are not well presented, and statistical methods are not well explained.

Results

Were all cases genetically confirmed?

This information should be added.

Page 4

Table 5

Typos error

101.6 instead of 101,6…

Page 5

Table 5

Typos error

…-1.2 (SD 5.1) instead of …-1.2 , SD 5.1…

Conclusion

Overall, the present findings have important implications for clinical practice in neurocognitive assessment of early treated PKU subjects.

Sometimes the inconsistent linkage between cognitive performances and metabolic control, support the hypothesis of an individual vulnerability to Phe.

Comments on the Quality of English Language

Minor editing of English language required

Author Response

Thank you very much for your detailed review.

All changes were made as suggested.

Typos were corrected.

The paragraph on blood samples was remade.

One child entered the study at age 5 ys but was tested at age 6ys (minimum age for the WISC). To avoid misunderstanding age range was adjusted.

All cases were genetically confirmed by mutation analysis of the PAH gene.

Reviewer 2 Report

Comments and Suggestions for Authors

Comments on the Quality of English Language

Minor editing of English language is needed

Author Response

Thank you very much for your detailed review.

Als changes were made as suggested.

Questions to be answered:

„Please explain the type of PKU included. Are all patients have classical PKU, or mild hyperphenylalaninemia. How were these patients diagnosed with PKU to start with.“ 

Patients were diagnosed with PKU according to their maximal blood Phe level before treatment, which had to be >600µmol/l to be classified as phenylketonuria. Patients with plasma Phe levels lower than 600 µmol/l were classified as mild hyperphenylalaninemia according to the German treatment guidelines and did not take part in the current study.

„Were patients treated with Kuvan, were also on classic diet as well?“ 

All patients receiving Kuvan were treated with a relaxed diet.

„What is the explanation of high blood Phe tolerance during therapy with Kuvan?“

Kuvan is increasing PAH activity because of its chaperone effect. This leads to a higher Phe- and protein tolerance in patients who are responsive to Kuvan.

„Was there dietary restriction in addition to Kuvan therapy?“ 

Patients on Kuvan were still on some degree of protein restriction, but the diet could be relaxed in all Kuvan patients.

“These patients had no classic PKU, were treated with Sapropterin dihydrochloride (KuvanTM), and showed a comparatively lower blood Phe tolerance before treatment”. What do you mean by “had no classic PKU”?“ 

Thank you for pointing to this paragraph. Indeed, it should better read: These patients had a moderate PKU, were treated with Sapropterin dihydrochloride (Kuvan TM) and ...

Yes, fluctuations were still within the target range.

Yes, Palynziq seems to have an even better effect on IQ than Kuvan. I tested some patients before they started with Palynziq and 6 to 12 months later. It was fascinating. However, the sample remained too small for presenting reliable results. I would like to encourage colleagues at metabolic centers to continue this research on the cognitive effects of Pegvaliase therapy.

Reviewer 3 Report

Comments and Suggestions for Authors

This is an interesting manuscript that brings knowledge to an important topic.

I suggest using the same units when referring to Phe values and commenting on how these determinations were made, that would allow us to better see the consequences of the treatments.

I also suggest complementing the presentation of results by adding graphs of the evolution of the values in time and treatment.

Author Response

Thank you very much for your suggestions. 

Units were adjusted.

Yes, results are presented somewhat complicated. Authors found that graphs may make things even more complicated.